# Using Live and Video Stimuli to Localize Face and Object Processing Regions of the Canine Brain

**DOI:** 10.3390/ani12010108

**Published:** 2022-01-04

**Authors:** Kirsten D. Gillette, Erin M. Phillips, Daniel D. Dilks, Gregory S. Berns

**Affiliations:** Psychology Department, Emory University, Atlanta, GA 30322, USA; kirsten.gillette@emory.edu (K.D.G.); erinmp@princeton.edu (E.M.P.); dilks@emory.edu (D.D.D.)

**Keywords:** fMRI, dogs, visual perception

## Abstract

**Simple Summary:**

We showed dogs and humans live-action stimuli (actors and objects) and videos of the same stimuli during fMRI to measure the equivalency of live and two-dimensional stimuli in the dog’s brain. We found that video stimuli were effective in defining face and object regions. However, the human fusiform face area and posterior superior temporal sulcus, and the analogous area in the dog brain, appeared to respond preferentially to live stimuli. In object regions, there was not a significantly different response between live and video stimuli.

**Abstract:**

Previous research to localize face areas in dogs’ brains has generally relied on static images or videos. However, most dogs do not naturally engage with two-dimensional images, raising the question of whether dogs perceive such images as representations of real faces and objects. To measure the equivalency of live and two-dimensional stimuli in the dog’s brain, during functional magnetic resonance imaging (fMRI) we presented dogs and humans with live-action stimuli (actors and objects) as well as videos of the same actors and objects. The dogs (*n* = 7) and humans (*n* = 5) were presented with 20 s blocks of faces and objects in random order. In dogs, we found significant areas of increased activation in the putative dog face area, and in humans, we found significant areas of increased activation in the fusiform face area to both live and video stimuli. In both dogs and humans, we found areas of significant activation in the posterior superior temporal sulcus (ectosylvian fissure in dogs) and the lateral occipital complex (entolateral gyrus in dogs) to both live and video stimuli. Of these regions of interest, only the area along the ectosylvian fissure in dogs showed significantly more activation to live faces than to video faces, whereas, in humans, both the fusiform face area and posterior superior temporal sulcus responded significantly more to live conditions than video conditions. However, using the video conditions alone, we were able to localize all regions of interest in both dogs and humans. Therefore, videos can be used to localize these regions of interest, though live conditions may be more salient.

## 1. Introduction

Previous imaging studies that have localized regions of the brain in both humans and canines have generally relied upon video and 2D image stimuli [1,2,3,4,5]. The use of video images is largely a matter of convenience for the experimenter, and although it may not make much difference for human neuroimaging studies, very little is known about how dogs perceive 2D images and whether they are perceived as referents for their real-world counterparts. The question of whether 2D images serve as valid stimuli depends on whether they have ecological validity for the subject. Humans readily equate images on a screen with their real-world counterparts, but there is scant evidence that dogs do. Previous studies have suggested dogs can abstract from iconic representations to their real-world counterparts, though this was not from images shown on a screen [6]. Other studies suggest that dogs’ brains show no significant difference between processing live faces versus portraits of faces [7]. In a recent dog fMRI study, our group found that a reward-system response could be associated with either objects or pictures of the same objects, but this reward response did not automatically transfer to the other condition. For example, a dog trained to associate a reward with a picture of an object did not show a reward response to the object itself [8]. This finding raises the question of whether live-action stimuli would be more ecologically valid for the study of faces as well as objects in dogs.

The idea of using live stimuli in neuroimaging studies, although not common, has some precedence in the human literature. A few studies have suggested that using 2D stimuli may not be equivalent to using 3D stimuli. For example, the human primary motor cortex responds more to live motor acts than videos of the same motor acts [9]. Additionally, motor corticospinal excitability increases in the arm muscle when watching live vs. video dance, which the authors theorize is the result of social cues present in the live condition [10]. As for object processing regions, there may even be different neural circuits for processing live rather than video stimuli, as repetition effects when processing videos of objects do not carry over to the same objects presented as live stimuli [11]. Additionally, the superior temporal sulcus is preferentially activated by moving stimuli rather than static stimuli, whereas the fusiform face area was not, which suggests that some face areas respond preferentially to more dynamic cues [12]. We theorize that live stimuli have attributes that video stimuli lack, such as social cues, depth information, multiple angles, more realistic motion, no frame rate effects, size cues, and animacy. Because we lack understanding of the dog’s visual system, these attributes may be particularly important in dogs, who are more sensitive to motion and frame rate effects on screens and have lower visual acuity compared to humans [13,14].

Because of these fundamental differences in the visual systems of dogs and humans, it is possible that their neural responses and consequent visual representations may be substantially different, especially to video stimuli. Using fMRI, we measured activation in cortical face and object areas in response to both live-action and video stimuli which consisted of human faces and objects. We opted to measure dogs’ responses to human faces rather than dog faces out of convenience, and prior studies suggest dogs respond to human facial cues [15,16,17]. If live-action stimuli are more salient to a certain brain region, then we would expect the activation in that region to be greater for the live condition rather than the video condition.

## 2. Materials and Methods

Participants were both dogs (*n* = 7) and humans (*n* = 5) from the Atlanta community. In total, 12 dogs participated in the experiment, though data were only retained for 7 of them. These seven dogs included 4 females and 3 males, all spayed/neutered, ranging in age from 3 to 11 years old. This sample included 1 boxer mix, 1 border collie, 1 Boston terrier mix, 2 lab-golden mixes, 1 golden retriever, and 1 pit mix. Prior to this experiment, all dogs had completed a training program that prepared them to be comfortable within the scanner environment and had participated in prior scan sessions (6–21). All dogs had previously demonstrated an ability to lie awake and unrestrained during scanning while viewing stimuli on a projection screen prior to this experiment. The human participants consisted of 2 females and 3 males ranging in age from approximately 21 to 30 years old, all of whom were right handed.

Stimuli consisted of blocks of live faces, live objects, video faces, and video objects. The live and video versions of stimuli showed the same objects and faces. The four objects were a pinwheel, a stuffed caterpillar without a face, a sandbox toy, and an aqua saucer pool toy (Figure 1). All were novel to the participants. The faces shown were those of lab members and other actors, who were all females of similar ages and were unfamiliar to the participants. During live runs, each of the four actors would stand in front of the participant in the scanner bore, one at a time, and make different facial expressions without making eye contact with the participant. A black curtain was hung behind the actors and over a table in front of the actors to match the scene showed in the videos. To show live objects, one actor would sit underneath the table, out of view, and hold the object up on a black stick in front of the participant. During video runs, a translucent screen was placed in front of the participants, and pre-recorded videos of the same moving stimuli were projected onto this screen for the participant to view. Dogs lay down in a sphinx position to watch stimuli, while humans were supine and viewed stimuli via a mirror attached to the head coil. Though humans watched through a mirror, the perceived image was 3D.

Each run consisted of four blocks of objects and four blocks of faces in a random order for a total of eight blocks per run. Each 20 s block showed either four faces or four objects for approximately 4–5 s each. Three runs consisted of live stimuli, and three runs consisted of video stimuli for a total of six runs. Though all runs of a condition (i.e., live or video) were presented successively, we randomized which condition a subject was presented with first. Owners stayed out of the dog’s sight except during interstimulus intervals that lasted approximately five seconds during which they could treat or praise their dogs as they felt was needed. However, we aimed to treat the dogs as little as possible to minimize motion throughout the scan. Stimulus onsets and offsets were recorded with a button box controlled by an experimenter seated next to the presentation area.

The scanning protocol for dogs in this study was the same as in previous studies [1]. All scans were obtained using a Siemens 3T Trio whole-body scanner. The dogs’ functional scans were obtained using a single-shot echo-planar imaging (EPI) sequence that acquired volumes of 22 sequential slices of 2.5 mm with a 20% gap (echo time (TE) = 25 ms, repetition time (TR) = 1260 ms, flip angle = 70°, 64 × 64 matrix, 2.5 mm in-plane voxel size). Approximately 1300 functional volumes were obtained for each dog over six runs. For dogs, slices were oriented dorsally to the brain with the phase-encoding direction right to left. For humans, axial slices were obtained with phase-encoding in the anterior–posterior direction. To allow for comparison with the dog scans (same TR/TE), multiband slice acquisition was used (Center for Magnetic Resonance Research, University of Minnesota) for the humans with a multiband acceleration factor of 2 (GeneRalized Autocalibrating Partially Parallel Acquisitions (GRAPPA) = 2, TE = 25 ms, TR = 1260 ms, flip angle = 55°, 88 × 88 matrix, 44 2.5 mm slices with a 20% gap). A T2-weighted structural image was also acquired for each dog and a T1-weighted magnetization prepared rapid gradient echo (MPRAGE) for each human.

Analysis of Functional NeuroImages (AFNI) (National Institutes of Health) was used to preprocess and analyze the functional data [18,19]. Preprocessing of the fMRI data included motion correction, censoring and normalization. Censoring was performed based on both signal intensity and motion. Volumes with either more than 1 mm of scan-to-scan movement or more than 1% of voxels flagged as outliers were censored from further analysis. To improve signal-to-noise ratio, the remaining data were spatially smoothed with a 6 mm Gaussian kernel. Additionally, a mask was drawn in functional space for each dog in the cerebellum, which was used to censor the data further by removing volumes where the beta values extracted from the cerebellum were assumed to be beyond the physiologic range of the blood-oxygen-level-dependent (BOLD) signal (|signal change| > 3%) for each trial. Of the twelve dogs that completed the study, seven had at least 66% of their data retained for both the live and video runs. This criterion was set so that there was ample reliable data to compare between the live and video conditions for each dog. Further statistical analysis focused on the humans and these seven dogs.

Task-related regressors for each experiment were modeled using AFNI’s dmUBLOCK and stim_times_IM functions and were as follows: (1) live faces; (2) live objects; (3) video faces; (4) video objects. This function created a column in the design matrix for each trial, allowing for the estimation of beta values for each trial. Data were censored for outliers as described above for the contrasts of interest. A series of contrasts were pre-planned to assess the main effects of faces versus objects and whether they differed between live and video versions. The contrast [all faces—all objects] was performed to identify regions that differentially respond to all faces versus all objects, independent of live or video conditions.

Regions of interest (ROIs) were defined from the contrast [all faces—all objects]. The ROIs for dogs in this study were the primary dog face area, as defined in previous dog fMRI studies [3,4], as well as the secondary dog face area along the ectosylvian gyrus which we believe to be analogous to human posterior superior temporal sulcus, and the lateral occipital complex along the entolateral gyrus. The ROIs for humans in this study were the fusiform face area, the posterior superior temporal sulcus, and the lateral occipital complex. To localize the ROIs for each dog and human, we overlaid the contrast of [all faces—all objects] onto each of individuals’ mean image in AFNI. We then varied the voxel threshold (*p* < 0.05) of the statistical map for each dog until one or two clusters near the ROI remained that were 10–40 voxels. For humans, we varied the voxel threshold (*p* < 0.05) of the statistical map until clusters remained that were 100–400 voxels in size since the human brain is approximately 10× larger than the average dog brain. In all subjects, we were able to localize the regions of interest, except for LOC in one dog participant. The ROI selection procedure is designed to identify the most likely clusters associated with the particular contrast. This is the same procedure we used in Aulet et al. (2019) [1]. It is based on the assumption that there are, in fact, face areas and object areas (for both dogs and humans) and merely aims to localize where they are. This assumption is based on both our prior results and others [3,4,20].

We then aimed to determine the relative contribution of live versus video and, secondarily, whether any such effects differed between dogs and humans. Using the masks created by the aforementioned ROIs, we used 3dmaskave to extract beta estimates for each trial in each ROI. This was performed using the ‘stim_times_IM’ option in 3dDeconvolve. To further decrease the effect of outliers due to motion, trials in which the absolute value of the beta estimate was greater than 4% from the implicit baseline were discarded from further analyses. Using SPSS 27 (IBM), we performed stepwise regression using the linear mixed-model procedure that minimized Akaike’s Information Criterion (AIC). These models included effects for species (dog, human), stimulus content (face, object), stimulus format (live, video), and ROI (primary face area, pSTS, LOC). Thus, even though the ROIs were defined by face vs. object, the contribution of this effect was factored out from the other terms.

## 3. Results

All of the main effects in the mixed-effects model were significant, as well as several key interactions (Table 1). As expected, face vs. object (FO) was significant because that is how the ROIs were localized (*p* = 0.006). The interaction of FO × ROI was significant because the LOC was localized by the opposite contrast, resulting in an opposite effect for that ROI (*p* < 0.001). With these effects factored out, we then found that live vs. video (LV) was significant (*p* = 0.036) and that this differed between dogs and humans because the interaction term, species × LV, was also significant (*p* = 0.037). The activation was generally larger in humans for all ROIs in all conditions and this effect was magnified in the video conditions, as seen in the bar graphs in Figure 2, Figure 3 and Figure 4. In other words, dogs had a greater decrement in activation to video stimuli than did humans.

## 4. Discussion

In summary, using both live and video stimuli, we localized the primary dog face area, a secondary dog face area, and the lateral object area in dogs, as well as the analogous regions in humans. In general, we found similar patterns of activation in both dogs and humans. For face stimuli, live conditions resulted in significantly greater activation than video conditions, which was more evident in dogs. In humans, secondary face regions, such as the posterior superior temporal sulcus, process increasingly social and dynamic aspects of faces [23,24,25]. Thus, it may be more efficient to localize this region using live-action stimuli rather than video stimuli because live-action stimuli can better capture dynamics than video stimuli. This effect may be especially pronounced in dogs because dogs may need context not present in video conditions to better perceive social aspects. Additionally, the dog visual system is more reliant on motion compared to that of humans, and live stimuli lack the potentially distracting effect of frame rate, to which dogs may be more sensitive than humans [13,14]. Thus, the face ROIs we examined in dogs activate in response to video stimuli but not as robustly.

Perhaps dogs’ visual responses are more variable than humans’. There are two primary sources of variability in brain imaging studies: (1) functional and (2) anatomical. Functional differences arise from heterogeneity in cognitive strategies for processing the study task. These tend to become more evident and problematic with complex tasks, in which a subject can arrive at a solution from different cognitive strategies. Our task did not involve any decision making on the subject’s part. As such, it was simply a passive task, where they watched the stimuli. Regions early in the visual processing stream should be relatively insensitive to differences in cognitive strategy as there is not much that can be done to alter the information coming from the retina. Anatomical differences are a different story. There are wider morphological differences in dogs than in humans. How much of this is breed related vs. individual heterogeneity? In our previous study with 50 service-dogs—all labrador/golden retriever crosses—we found size variations up to 30% [22]. This is perhaps the most compelling reason to analyze the data in individual space and use an atlas for visualization only. Due to the morphological differences between not only breeds, but also within breeds, analysis in individual space ensures we do not miss areas that may be lost in a group analysis. Other experiments that did not find a dog face area analyzed their results in a group space, potentially missing these face areas by averaging brains together [2,7]. For this reason, face areas in the human literature tend to be analyzed in individual space and are defined functionally, not anatomically. We believe this should be the same approach to dog data, especially given the greater differences between individual dogs.

As for functional variability, even within humans, there is enough variability in the BOLD response that early investigators were concerned about its effect on statistical results [26]. Moreover, it is also likely that regional differences within the brain may exist, but because of the difficulty in precisely stimulating activation outside of sensory regions, it is not feasible to determine the hemodynamic response function (HRF) throughout the cortex. Subcortical structures may have slightly different response functions, too, but it is not straightforward to determine how they are different. Again, these issues, which may initially appear to be species related, are actually a source of subject variation within species [27]. We are not aware of any data that convincingly demonstrates that there are further differences at the species level. In the end, the variation in HRF is small and does not appear to affect the results, at least for false positives. There may be a small effect in false negatives (failing to detect activation when it is actually present), but this would be more likely to occur in rapid event-related designs where timing is critical, not in block designs as used in our study.

Ever since we began dog fMRI, we noted that all dogs have periods of excessive movement [28]. How much is too much? That depends on the design of the experiment and whether one is analyzing for task-related activity or resting state. Resting state is much more sensitive to movement, where anything more than approximately 0.2 mm can affect results [29,30,31]. Larger movements can be tolerated in task-related fMRI, up to a point, generally the size of a voxel. Motion correction deals with the movement of the image, but spin-history effects are nonlinear and cannot be completely regressed out. There is ample evidence in the human literature that censoring out (also called ‘scrubbing’) high-motion data points improves statistical validity [32]. There are many techniques for accomplishing this, including independent component analysis (ICA), spike regression, and scrubbing/censoring. We know that the maximum BOLD response is approximately 5% and that only occurs in primary sensory/motor regions, so the observation of a BOLD response of that size in a region unrelated to the task is almost certainly a spurious result that should not be included in the statistical analysis. The ventricles would be an ideal location for such an ROI, but because they are small, we use the cerebellum, which is not expected to be involved in any of the tasks we use.

There are several limitations to our study. For example, in the live conditions, the stimuli entered the view of the dog, but in the video conditions, the stimuli were already in frame. Therefore, the live conditions may have had more motion than the video conditions even though logging of the trial did not begin until the actor or object was already in view of the dog. However, there is also more motion preceding the object condition, as the objects also entered from out of view, so whatever effect is associated with this difference in movement, it is likely to be similar in both the face and object conditions and therefore not contribute to any interaction. Additionally, this study had a small number of participants, both dogs and humans. Future research could replicate these results using different actors and objects with greater sample sizes. Furthermore, we did not control for the faces of conspecifics or the species of faces. Prior research suggests that visual areas in dogs may be species specific, activating preferentially to the faces of conspecifics rather than faces in general [2]. Additionally, background stimuli present in the live condition, such as odor, may not have been controlled for in the video stimuli.

An evolutionary perspective of emotion suggests that its expression serves as an information signal between sender and receiver and that there are commonalities across mammals [33,34]. In humans and other primates, facial expression is a core manifestation of emotion, and they have evolved facial musculature for this purpose [35,36]. Dogs are evolutionarily distant from primates, but through domestication may have acquired some ability to receive emotional signals from humans. The question, then, is through what channel? We can look at the dog’s expressive capacity as a starting point. What a dog can express emotionally is a good bet to be a salient signal for which they also have receptive capacity. Growling, for example, is recognized by other dogs as a warning signal, but so, too, are facial expressions. Humans and dogs can recognize the difference between a dog’s submissive grin, a happy relaxed smile, and a bearing of teeth [37]. For this to occur, dogs should possess neural circuitry that can process faces. In primates, we can identify a series of face-responsive regions, including the fusiform face area (FFA), occipital face area (OFA), and posterior superior temporal sulcus (pSTS), but the amygdala is more often implicated in the processing of emotional expression [38]. The vast majority of the neuroscience literature of face processing, however, was obtained with either static pictures or video stimuli. Although these evoke responses in humans, it has not been clear whether these types of stimuli are appropriate for dogs. Here, we find that video stimuli are sufficient to elicit activity in the basic face-processing circuits of the dog, but that live stimuli result in a significant boost. This difference may become more important when probing the neural circuitry of emotional processing, which likely depends on subtle dynamics of microexpressions.

## 5. Conclusions

Overall, we found that video stimuli were sufficient in defining our ROIs. However, in face regions in both dogs and humans, the live condition yielded greater activation within these regions than the video condition. This effect may be especially pronounced in regions in dogs that respond to dynamic aspects of stimuli, such as the secondary dog face area. In the future, more fMRI studies, including those outside of face and object processing, could be performed with live stimuli that better represent stimuli encountered naturally in the world. This would allow us to observe how the brain processes more naturalistic stimuli and potentially improve upon our ability to localize regions of interest using fMRI.

## Figures and Tables

**Figure 1 animals-12-00108-f001:**
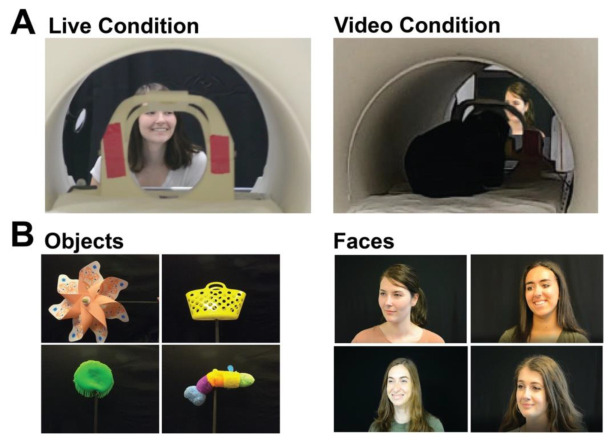
Presentation format and stimuli. (**A**) Examples of what dogs and humans saw when viewing live (left) and video (right) conditions (note image of the same actor is projected on a screen). (**B**) Stimuli examples of objects (left) and faces (right).

**Figure 2 animals-12-00108-f002:**
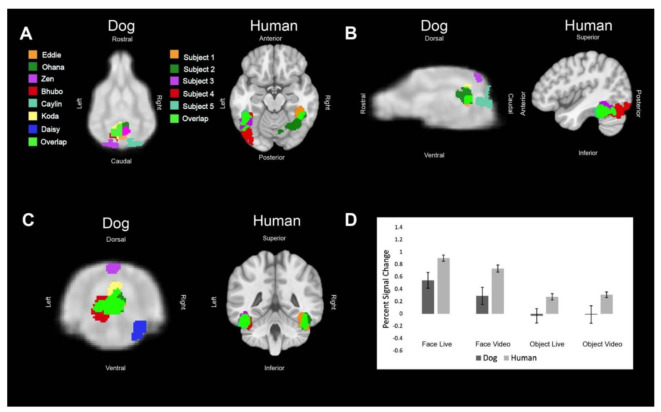
Definition and activations within the primary dog face area and human fusiform face area. For visualization purposes, these regions of interest have been spatially normalized and overlaid on to their respective atlases (humans: Montreal Neurological Institute atlas [21]; dogs: CCI atlas [22]). Each color represents the ROI of one dog or human or an area where the regions of interest overlapped. (**A**) Dorsal (dog) and axial (human) views of individual ROIs. (**B**) Sagittal views. (**C**) Transverse (dog) and coronal (human) views. (**D**) The bar graph shows the average percent signal change for each species for each condition relative to the implicit baseline. Error bars are the standard error.

**Figure 3 animals-12-00108-f003:**
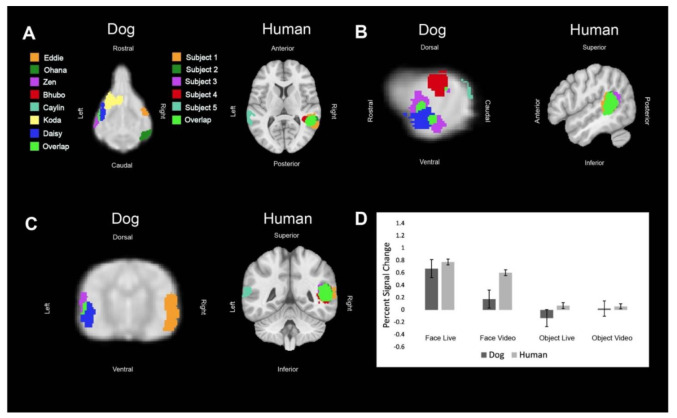
Definition and activations within the human posterior superior temporal sulcus and its analog in dogs. Each color represents the ROI of one dog or human or an area where the regions of interest overlapped. (**A**) Dorsal (dog) and axial (human) views of individual ROIs. (**B**) Sagittal views. (**C**) Transverse (dog) and coronal (human) views. (**D**) The bar graph shows the average percent signal change for each species for each condition relative to the implicit baseline. Error bars are the standard error.

**Figure 4 animals-12-00108-f004:**
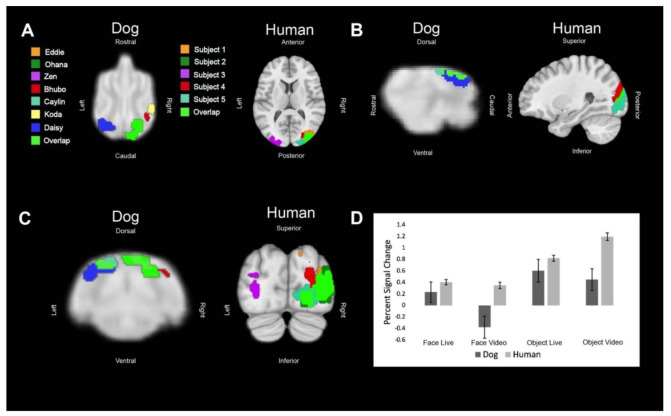
Definition and activations within the human lateral occipital complex and its analog in dogs. Each color represents the ROI of one dog or human or an area where the regions of interest overlapped. (**A**) Dorsal (dog) and axial (human) views of individual ROIs. (**B**) Sagittal views. (**C**) Transverse (dog) and coronal (human) views. (**D**) The bar graph shows the average percent signal change for each species for each condition relative to the implicit baseline. Error bars are the standard error.

**Table 1 animals-12-00108-t001:** Results of the mixed-model analysis of the ROIs.

Effect	Numerator df	Denominator df	F	Significance
Species	1	1592	45.47	<0.001
Face/Object (FO)	1	1592	7.43	0.006
Live/Video (LV)	1	1592	4.40	0.036
ROI	2	1592	3.89	0.021
FO × LV	1	1592	13.98	<0.001
FO × ROI	2	1592	55.50	<0.001
Species × LV	1	1592	4.35	0.037

## Data Availability

All study data, including videos, fMRI data, and code used for analysis, are available upon request.

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
