# Peer review of "Using Live and Video Stimuli to Localize Face and Object Processing Regions of the Canine Brain"

_animals, 2022, doi:10.3390/ani12010108_

Round 1

Reviewer 1 Report

Manuscript provides an update on comparative use of visual stimuli in fMRI and behavioral studies based on modality of presentation, live versus video. Overall well written manuscript with minor edits and clarifications requested.

Line 92: provides description of human "live" visual stimuli setup through use of a mirror. This variation in the live visual stimuli presentation setups between the dog and human may not represent a relatively equal comparison as the dog's live visual stimuli setup provided a direct line of sight thereby limiting the potential distortions or altered perceptions that may be presented with the use of a mirror. How did you account for this limitation in setup between the two species in your interpretation? Would this of had a potential to impact your interpretation within a species or your interpretation across the two species? 

Line 99: Please clarify if the live stimuli runs preceded the video stimuli runs or if the run order was randomized between live and video. If order was set and live or video were consistently presented first, this may suggest the  need to address the potential for a bias in the data based on familiarity or recall moving from one modality to the next as all objects and person's were reported as novel on initial presentation.  

Line 178: Please insert in-text reference for Figure 3.

Line 190: Please insert in-text reference for Figure 4. 

Figure 2, 3 and 4: Images would benefit from clear labeling between dog and human brain for a more general audience. 

Line 251: Considerations for other cross-modal background stimuli that are introduced in the live stimuli and may not necessarily have been controlled for in the video stimuli (e.g. odor) may also represent a limitation, especially when evaluating one group with inter-species examples and another with intra-species examples. 

Reviewer 2 Report

In this recent fMRI study dogs and humans were presented with live action and video stimuli (both faces and objects). I think the topic is interesting, however, the method design and the sample sizes are not adequate to properly test the hypothesis presented by the authors. In the following, I will mention my main concerns and comments.

Introduction

I found the introduction short. The authors do not mention dog behaviour studies where e.g. pictures and videos or real life stimuli and videos were compared. Maybe not much studies have been done on this topic, but there are a few that should have been mentioned in more detail, e.g.:

Huber, L., Racca, A., Scaf, B., Virányi, Z., and Range, F. (2013). Discrimination of familiar human faces in dogs (Canis familiaris). Learn. Motivat. 44, 258–269. doi: 10.1016/j.lmot.2013.04.005

The examples of human studies where live vs video stimuli were presented are not static but motor acts. The increased activation of the brain reported in these studies might be due to the fact that motor action elicits more activation in the brain if we watch a live action. But what about static stimuli? Because here in the present study static stimuli are presented, if I understood correctly.

What about other species? Face sensitivity of primates are not mentioned (e.g. Burke, D., and Sulikowski, D. (2013). The evolution of holistic processing of faces. Front. Psychol. 4:11. doi: 10.3389/fpsyg.2013.00011)

I further miss information about dog fMRI studies. A paragraph should mention the relevance of these studies (being non-invasive, etc.). Also, why would human faces be more relevant for dogs than other dog faces? I am aware of the importance of human facial cues in dog-human communication but it is not mentioned in the introduction, e.g.:

Müller, C. A., Schmitt, K., Barber, A. L., and Huber, L. (2015). Dogs can discriminate emotional expressions of human faces. Curr. Biol. 25, 601–605. doi: 10.1016/j.cub.2014.12.055

Nagasawa, M., Murai, K., Mogi, K., and Kikusui, T. (2011). Dogs can discriminate human smiling faces from blank expressions. Anim. Cogn. 14, 525–533. doi: 10.1007/s10071-011-0386-5

Gácsi, M., Miklód, Á., Varga, O., Topál, J., and Csányi, V. (2004). Are readers of our face readers of our minds? Dogs (Canis familiaris) show situation-dependent recognition of human's attention. Anim. Cogn. 7, 144–153. doi: 10.1007/s10071-003-0205-8

In the intro, an important dog fMRI study is not mentioned where live faces vs 2D pictures were used. They found no differences between the portrait (photo) and the live face presentations:

Szabó D, Gábor A, Gácsi M, Faragó T, Kubinyi E, Miklósi Á and Andics A (2020) On the Face of It: No Differential Sensitivity to Internal Facial Features in the Dog Brain. Front. Behav. Neurosci. 14:25. doi: 10.3389/fnbeh.2020.00025

Methods

I believe that for statistical analyses the five human and seven dog data are not enough. I encourage the authors to gather more data (in the abstract N=12 dogs are written, although in the methods only N=7 dogs; it was misleading).

Furthermore, demographic data of dogs and ages of human participants are not mentioned.

The authors used multiple pair sample t tests (I think GLMs would have been a better choice as in the model both species and type of stimuli could have been included) but did not correct for multiple comparisons (at least it is not mentioned). That is important to exclude false positive results (especially if the sample size is that small).

Discussion

Another important and major limitation of the study is only mentioned in the discussion, however, it should have been included in Methods. 

“in the live face condition, the actors walked into view of the dog, but in the video face condition, the actors were already in frame. Therefore, the live face condition may have had more motion than the video face condition”

That is an important difference. Since the two conditions are not identical, thus we can not know whether the observed differences are present due to more motion in the live condition or because of the different processes of video vs live stimuli.

Also, the authors wrote:

“Additionally, this study had a small number of participants, both dogs and humans. Future research could replicate these results using different actors and objects with greater sample sizes”

I don't see the relevance of replicating the same study design with greater sample sizes. Then what would this study give to the scientific audience if the results are not reliable and it needs replication?

Summary

I believe the current paper needs major improvements. The sample sizes are not great enough to perform statistical analysis and to have valid results. Furthermore, due to the difference between video and live conditions, it can not be reliably compared. We will not know whether the received results are due to the different processes of the two stimuli, or simply because of the more movements in the live condition.

Reviewer 3 Report

This paper reports fMRI evidence of how dogs visually process live objects and people compared to images of objects and people. Humans were also studied as a inter-species comparison. I really enjoyed reading this ms and I think it will make a substantial contribution to the field of dog cognition, which relies heavily on visual stimuli, often 2D images, in behavioural tests. I just have a few suggestions for further improvements.

L42-45 were unclear. I couldn't make sense of why the second half of the sentence is different from the first part. Please clarify this. It may need another sentence or two.

L63-64. The Miller and Murphy review is excellent, but there has been another, recent review with more information about various aspects of visual processing in dogs. This might further justify the current study because it highlights how little is known - still - about how dogs see the world: https://link.springer.com/article/10.3758/s13423-017-1404-7

L73 - what were the dog breeds/sex/age for the 7 included dogs? This could be relevant given the difference in visual anatomical structures in different breeds.

Also - mention here that 12 dogs participated but data were only retained from 7 of them. it's mentioned below but should also be noted here.

Were the human participants all same-handed? I ask because it looks like there's more right hemispheric activity in Fig 2 for some human participants, compared to the left hemispheric activity that is more common. Is this to do with handedness or just one of those flukey things that you can get with these technologies? Either way, handedness should be noted in the participants portion of the methods section, and perhaps the reason why the hemispheric differences are seen should be mentioned in the discussion.

L176 - suggest moving Fig 2 here, just below where it is referred to in text.

Figs 3 and 4 are not referred to anywhere in text.

Fig 4 - graph. I understand that the differences are negative, rather than positive as in the other figures, but it looks odd. Consider flipping it, but it's not a dealbreaker if it needs to stay as is. Perhaps, in that case, it could be briefly explained in the Fig 4 caption that it looks upside down because of the negative change.

Round 2

Reviewer 2 Report

I agree that most fMRI (and even EEG) studies are conducted with small sample sizes. I also understand that training these dogs are very time consuming. However, in recent fMRI studies at least eleven, twelve dogs are included and even more human participants. For example, in the last dog-human fMRI study N=20 dog and N=30 human participants were included.

Bunford, N.; Hernández-Pérez, R.; Farkas, E.B.; Cuaya, L.V.; Szabó, D.; Szabó, Á .G.; Gácsi, M.; Miklósi, Á .; Andics, A. Comparative Brain Imaging Reveals Analogous and Divergent Patterns of Species and Face Sensitivity in Humans and Dogs. The Journal of Neuroscience 2020, 40, 8396-8408, doi:10.1523/jneurosci.2800-19.2020.

The authors wrote:

“Our original fMRI studies in dogs were all small samples, but the results have been largely replicated by other groups, thereby increasing confidence in the validity of the methods as well as the findings.”

I think this is important and could have been included in discussion (with references). However, a recent study (Bunford, N.; Hernández-Pérez et al., 2020) with greater sample size could not support previous findings (on dog face area) which were conducted on smaller sample sizes (Dilks et al., 2015; Cuaya et al., 2016).

With the inclusion of dog breeds, another important factor needs to be considered. Difference in visual anatomical structures in different breeds. As two brachycephalic dogs participated, this should have been mentioned.

With the current sample size and without FDR correction, I think this paper still needs major improvements.